

# Compounds from medicinal plants produced in hairy root and transgenic hairy root cultures: a review

Araceli Urquiza-López[1], Crescencio Bazaldúa[2], Alexandre T. Cardoso-Taketa[1] and Maria Luisa Villarreal[1]

[1] Centro de Investigación en Biotecnología, Universidad Autónoma del Estado de Morelos, Cuernavaca, Morelos, Mexico
[2] Centro de Desarrollo de Productos Bióticos (CEPROBI-IPN), Yautepec, Morelos, Mexico

Corresponding authors
Crescencio Bazaldúa,
cbazaldua@ipn.mx
Alexandre T. Cardoso-Taketa,
ataketa@uaem.mx

## ABSTRACT

**Background**. Plants produce a wide variety of molecules, and some of them are of medicinal importance. Biotechnological systems such as suspended cells and hairy roots (HR) are used to produce plant compounds in a stable and controlled manner. HRs are obtained through a genetic transformation mediated by *Rhizobium rhizogenes* (*R. rhizogenes*), a Gram-negative bacterium that randomly inserts the transfer DNA (T-DNA) from the root-inducing plasmid (pRI) into the chromosome of a plant cell. Gene expression of T-DNA in plant cells induces a metabolic change to promote HR syndrome. The primary benefits of producing medicinal plant compounds in these biotechnology systems include the large yield of organ-specific bioactive compounds, production of *de novo* secondary metabolites, and scaling up to bioreactors for the industrial production of medicinal plant compounds. This article aims to describe the applications and advantages of the biotechnological system of hairy root cultures, which is used to obtain specific or *de novo* compounds of plant secondary metabolism in the last fourteen years.

**Methodology**. A broad bibliographic search was accomplished of articles that report the HR cultures to produce bioactive compounds of medicinal plants. To find the largest number of reports in the past fourteen years, different databases for web searchers, such as Scopus, PubMed, Google Scholar, Web of Science, Redalyc, and SciELO were used. The reports mentioned here were collected and selected to include those that were of great relevance.

**Results**. One hundred and twelve research articles were selected to integrate this review. We highlight the principal advantages of hairy root cultures to produce secondary metabolites which are used as medicinal compounds. Hairy root cultures can produce a wide variety of organo-specific compounds, including *de novo* secondary metabolites, by activating complex metabolic pathways. This process is facilitated by the expression of *rol* genes which are inserted into the chromosome of the plant cell during genetic transformation mediated by *R. rhizogenes*. Therefore, stable and continuous molecules production over several years, as well as a higher yield than that in wild plants, have been observed. Another advantage is the easy scaling up into a bioreactor system.

**Conclusions**. Different plant species have successfully produced HR. The root cells in HR cultures have a complicated endomembrane system that facilitates the production of a wide variety of bioactive molecules of the secondary metabolism, such as phenols, terpenes, and alkaloids. Due to the random insertion of the pRI *rol* genes, different

secondary metabolism pathways are activated in each induced HR. Only a few HR lines synthesize higher concentrations of specific molecules found in the wild plant or *de novo* metabolites, including those used as medicinal compounds; and some of them are secreted into the culture medium.

# INTRODUCTION

Plants are the main source of the production of bioactive molecules that are used in medicine to treat diseases (*Atanasov et al., 2015*). Plant secondary metabolic pathways synthesized many compounds of pharmacological interest, either due to intrinsic factors related to the stage of phenological development or external factors such as the interaction with pathogenic organisms, predators, pollinators, and environment changes (*Tian, 2015*). Other specific factors are growth conditions and the accumulation of tissue-specific compounds, such as the accumulation of taxol in the bark of *Taxus brevifolia* Nutt. trees (*Sykłowska-Baranek et al., 2022*), the artemisinin in trichomes of *Artemisia annua* L. (*Dhankhar et al., 2022*), and the accumulation of alkaloids in idioblasts in *Catharanthus roseus* L. (*Roze, Chanda & Linz, 2011*).

The synthesis of medicinal compounds depends on highly variable factors, so their production is neither continuous nor uniform. The extraction processes from the wild plant yield a large amount of primary and secondary metabolites, but typically result in low yields of medicinal compounds, usually secondary metabolites. Therefore, large amounts of plant material are required, which may put medicinal plant species at risk of extinction due to indiscriminate use and overexploitation (*Rawat et al., 2019*). Moreover, complex purification techniques are necessary to extract specific active molecules. Because of these reasons, biotechnological systems like hairy roots (HR) and transgenic hairy roots (THR) and suspended cells are used to produce medicinal plant compounds. However, HRs have greater genetic and biochemical stability than cell suspension cultures (*Ochoa-Villarreal et al., 2016*; *Stiles & Liu, 2013*). This review describes the secondary metabolite production by hairy root (HR) as well as by transgenic hairy root (THR) cultures, where HR refers to those roots induced by wild *R. rhizobium* strains. On the other hand, THR are those induced by strains with a modified Ri plasmid (*Meng et al., 2019*).

Moreover, hairy roots have specialized cells with a complex endomembrane system, which facilitates the obtaining of organo-specific compounds (*Yoo et al., 2014*). The hairy roots are a result of genetic transformation mediated by *R. rhizogenes* (*Gutierrez-Valdes et al., 2020*). During the indirect transformation mechanism mediated by *R. rhizogenes*, the transfer DNA (T-DNA) from the bacteria root-inducing plasmid (pRI) is randomly inserted into the chromosome of the plant cell (*Makhzoum et al., 2013*). The T-DNA contains the *rol* genes, which, when fully inserted into the plant cell chromosome and expressed, induce the hairy roots phenotype.

The activation of some secondary metabolism pathways depends on the insertion site that promotes the production of specific compounds (*Bahramnejad et al., 2019*; *Desmet et al., 2020*). Each infected plant cell can produce a hairy root line that accumulates a certain type of molecule. In some cases, this super-producing line can synthesize compounds or *de novo* secondary metabolites by the activation of complex metabolic pathways (*Bazaldúa et al., 2014*; *Moreno-Anzúrez et al., 2017*; *Sarkar et al., 2018*). HR cultures serve as a versatile system capable of producing a wide range of plant medicinal compounds. The industrial production of bioactive plant molecules is carried out in HR bioreactor systems that work under controlled conditions. These systems can be optimized for compound production using elicitors or changes in the configuration and operation. This allows for the continuous obtention of active compounds and the preservation of wild plants (*Stiles & Liu, 2013*).

To guarantee an all-inclusive and unbiased coverage of the literature, we focus on the most recent research articles, published between 2011 and 2024, that detail the production of medicinal plant compounds using HR cultures. The number of articles (an average of sixty-two per year, PubMed) related to this research area in the past 14 years confirms the growing interest in HR cultures for obtaining bioactive compounds nowadays.

This review will contribute to clarifying and actualizing the knowledge of researchers interested in the HR culture systems. The goal of this review is to describe the applications and emphasize the advantages of the use of this system for obtaining different molecules of pharmacological interest or *de novo* compounds produced in specialized plant secondary metabolic pathways in plants.

## SURVEY METHODOLOGY

Research articles were collected using a systematic review in Google Scholar, Web of Science, Scopus, PubMed, Redalyc, and SciELO. The search terms used in this search were: "hairy roots", "medicinal plant compounds", "medicinal compounds produced in hairy roots", "hairy roots of medicinal plants", and "plant transformation by *Agrobacterium rhizogenes*". The search information was carefully separated according to relevance. This manuscript specifically analyzed the last 14 years of reports on the production of medicinal plant compounds by hairy roots. This review aimed to exclude publications that solely discussed obtaining hairy roots without disclosing the bioactive molecules generated in this biotechnological system.

### Generalities of *Rhizobium rhizogenes* (previously named *Agrobacterium rhizogenes*)

*R. rhizogenes* is a soil Gram-negative bacterium that causes The hairy roots (HR) syndrome (*Rawat et al., 2019*). The classification of the strains is based on the opines they produce and use as a source of carbon and nitrogen. In the cucumopine, mannopine, and mikimopine strains, the T-DNA is spread out in a specific area within the root inductor plasmid (pRI) (*Desmet et al., 2020*). However, in the agropine strain, the T-DNA is split into two parts inside the plasmid. These parts are called TL-DNA and TR-DNA, and there is a small piece of DNA called T-Central DNA (TC-DNA) between them. The plant cell does not receive this insert (*Ozyigit, Dogan & Artam Tarhan, 2013*). The root induction plasmid (pRi) is shaped

by virulence genes (*vir*), vegetative replication origin (*ori* V), conjugal transference genes (*tra*), the DNA of transference (T-DNA) region that contains the *rol* genes (responsible for HR syndrome), *aux* genes (generate the auxins synthesis, only agropine strains), *mas* genes (codify the mannopine synthase enzyme), and *ags* genes (responsible for agropine synthesis, only agropine strains) (*Desmet et al., 2020*; *Rawat et al., 2019*).

## Plant cell transformation mechanism mediated by *R. rhizogenes*

The indirect transformation mechanism in plant cells mediated by *R. rhizogenes* has been described in detail by *Chandra (2012)* and is shown in Fig. S1. Briefly, the interaction and insertion of the T-DNA bacteria into the chromosome of the plant cell are described. The infection mechanism of *R. rhizogenes* begins with the secretion of phenolic compounds and sugars by the plant cell because its cell wall and membrane have been damaged. The bacteria recognize the compounds and are attracted by chemotaxis to the plant cells. The transmembrane monosaccharide transport protein ChvE in the bacteria serves as a periplasmic sugar-binding protein that recognizes sugars and promotes the activation-expression of the chromosomal virulence genes *chv*A and *chv*B. The *chv*B genes code for a protein that participates in the formation of the β-1,2 cycloglucan, which allows the bacteria to anchor itself to the plant cell wall.

Phenolic compounds such as actetosyringone or dihydroxyacetosyringone are detected by the bacteria's transmembrane protein VirA, which are part of a two-component signal translation system (*Chandra, 2012*; *Krenek et al., 2015*). This protein is autophosphorylated and subsequently transphosphorylated to VirG, whose function is to activate or increase the transcription level of the *vir* genes found in pRI (*Krenek et al., 2015*). The VirD1-2 protein allows the separation of the double strand of DNA in the T-DNA region of the plasmid, leading to the synthesis of a new TDNA strand. Then VirD2 binds to the 5′ end and VirD1 binds to the 3′ end of the newly formed strand, allowing it to separate from the plasmid. Next, VirE protein binds to the formed strand to protect the T-DNA, generating the mature T-DNA, like the single-stranded DNA-binding proteins (SSBs). While VirB proteins form a pili that spans the plant cell wall and membrane. The mature T-DNA then moves through the pili and into the plant cell cytosol. VirF proteins also travel along the pili with the mature T-DNA. Once the T-DNA gets into the plant cell cytosol, the importins recognize the VirE2-3 protein and guide the T-DNA to the chromosome in the nucleus; subsequently, the VirF protein releases all the proteins that protect the T-DNA and randomly integrates it into the plant chromosome in an active transcription region. The plant cell then synthesizes the complementary strand of the inserted T-DNA.

Hairy root syndrome begins when the *rol*-genes contained in the exogenous T-DNA inserted into the plant cell chromosome are expressed, which induces a metabolic, morphological, and phenotypic changes (*Desmet et al., 2020*). The *rol* genes (A-D) will induce hairy roots in the plant, but each one develops different and specific activity in the plant physiology. The *rol*A negatively regulates the synthesis of some phytohormones such as kinetin, ethylene, abscisic acid, and gibberellin; *rol*B is the main inducer of HRs, acting as a tyrosine phosphatase that stimulates their development, and also promotes sensitivity to auxins and stimulates the activation of secondary metabolism in plants; *rol*C

acts by releasing cytosine conjugates and activates the production of alkaloids; together with *rol*A and *rol*B, they stimulate the generation of HRs and increase the production of secondary metabolites; *rol*D encodes an ornithine cyclodeaminase that generates an increase in resistance to pathogens (*Bahramnejad et al., 2019*; *Mauro & Bettini, 2021*).

All the *R. rhizogenes* strains contain a copy of the *rol*A, *rol*C, and *rol*D genes, but regarding *rol*B, agropine-type strains have an extra copy that is located in the TR-DNA section. This section also contains the *aux*1 and *aux*2 genes that in turn help in the development of HRs (*Bahramnejad et al., 2019*; *Desmet et al., 2020*). Agropine-type strains process the TL-DNA and TR-DNA sections independently, enabling them to generate multiple HR lines from a single explant (*Ozyigit, Dogan & Artam Tarhan, 2013*). The truncated T-DNA insertion has also been reported. When the plant cells recognize mature T-DNA as exogenous DNA, they activate their defense mechanisms and partially degrade the T-DNA. This scenario could potentially lead to the independent insertion of TL-DNA or TR-DNA, or partial insertion of each, resulting in abnormal roots or cell mass growth (*Desmet et al., 2020*; *Gelvin, 2010*; *Yousefian, Golkar & Mirjalili, 2021*).

## Infection mechanism for hairy root obtention

There are different techniques to induce HR in medicinal plants. Among the most used methods, the scalpel wound infection procedure, infection by inoculum, and co-culture infection are described. The scalpel wound technique consists of collecting the active *R. rhizogenes* strain with the tip of a scalpel and immediately making a cut in the plant explant at meristem sites, thus bringing the bacteria into direct contact only at the wound site of the plant (*Yousefian, Golkar & Mirjalili, 2021*). In the inoculum method, the *R. rhizogenes* strain is previously cultured in liquid medium to have an inoculum with an $OD_{600} = 0.8$. Then, three µL of the inoculum is injected in duplicate with a syringe into different meristems of each explant (*Amini, Fattahi & Nazemiyeh, 2024*).

Co-culture is another method. To perform this, an inoculum with an $OD_{600} = 0.6-0.8$ of the previously cultured *R. rhizogenes* strain is prepared. A wound is made in the explants with a sterile scalpel or an ultrasound bath is used for 45–80 s to generate damage in the cells of the explant. Then, the wounded explants are placed in the liquid medium together with the bacteria for 10–15 min. The explants are dried with sterile filter paper to remove the inoculum and are cultured in *in vitro* plant culture medium (*Amini, Fattahi & Nazemiyeh, 2024*; *Sandhya & Giri, 2022*). All the procedures include several antibiotic washes and the addition of the antibiotic to the culture medium to place the infected explants, which facilitate the elimination of bacteria and assure the growth of axenic HR.

## Bioactive molecules produced in hairy roots

Nowadays, interest in HR cultures is increasing and has been promoted as one of the main biotechnological systems for obtaining secondary metabolites with medicinal use (*Tian, 2015*; *Sonkar, Shukla & Misra, 2023*). Differentiated plant organs like HR have more complex metabolic pathways and can produce compounds more effectively than dedifferentiated plant cells (*Sonkar, Shukla & Misra, 2023*). Genetic transformation mediated by *R. rhizogenes* has been used to generate hairy roots in various plant species

(*Roychowdhury, Majumder & Jha, 2013*). Different *R. rhizogenes* strains, wild (only with pRI plasmid) and transformed (that have other plasmid, including the pRI), are used in the genetic transformation of plant cells. The transformed strains are modified with binary vectors that have sequences of reporter genes or specific constructions of recombinant DNA to increase the enzymes that intensify a metabolic pathway, producing specific bioactive secondary compound. The hairy roots induced with transformed *R. rhizogenes* strains are named transgenic hairy roots (THR) (*Bahramnejad et al., 2019*; *Bazaldúa et al., 2019*; *Gutierrez-Valdes et al., 2020*; *Ni et al., 2011*).

According to the above, *Park et al. (2011)* induced the THR in leaves and stems of *Scutellaria baicalensis* Georgi using the R1000 strain of *R. rhizogenes*. This strain has a plasmid with a recombinant DNA gene for the chalcone synthase enzyme. They observed that the obtained transgenic hairy roots produced more flavonoids, such as baicalein at 12.6 μg/mg DW and wogonin at 5.38 μg/mg DW, which is 2.85 and 8.82 times more than the wild plant roots.

In a different study, *Jin et al. (2021)* induced transgenic hairy roots in *Panax ginseng* C.A. Meyer with the R1601 strain of *R. rhizogenes*. This strain contains a plasmid with the *PAP1* gene (production of anthocyanin pigment 1) that was up-expressed in ginseng THR. The results showed that the production of phenolic compounds, including anthocyanins, increased until it was twice as high as the control group that did not have the gene.

On the other hand, *Wang et al. (2021)* used both wild and transformed *R. rhizogenes* ATCC 15834 strains to induce HR and THR in *Glycyrrhiza uralensis* Fisch. They transformed the strain of *R. rhizogenes* with three different plasmids containing genes to overexpress and to silence the biosynthesis pathway involved in the glycyrrhizin synthesis: the β-amyrin synthase gene (β-AS), the UDP-galactose/glucose-4-epimerase gene (UGE), and the auxin-responsive protein IAA gene (ARPI). They reported that the THR-UGE+ and THR-β-AS+ increased the production of glycyrrhizin to 2.9 and 1.7 times more than the HR induced by the ATCC15834 wild-type strain (*Wang et al., 2021*).

*Wu et al. (2022)* established the THR culture of *Glycyrrhiza inflata* Bat. with the R100 strain of *R. rhizogenes*. This strain possesses a plasmid with the AtMYB12 gene, which regulates flavonoid biosynthesis. They observed that the transgenic hairy root lines obtained accumulate licochalcone A and echinatin (2 and 4 times more, respectively) than the strain of *R. rhizogenes* without the AtMYB12 gene.

The efficiency of transformation is attributed to the development of hairy roots in the infected explant, which is associated with the *R. rhizogenes* strain used, the explant type, its developmental stage, the plant species, and the plant organ (*Danphitsanuparn et al., 2012*; *Thiruvengadam et al., 2014a*; *Thwe et al., 2016*).

According to the above, *Tavassoli & Safipour Afshar (2018)* induced the HR culture of *Althaea officinalis* L. They reported that the hairy roots obtained from the A4 strain of *R. rhizogenes* were the major producers of flavonoids (three times more than the wild roots), while the A13 strain showed the most accumulation of other phenolic compounds (4.7 times more than the wild roots).

*Miao et al. (2021)* established HR cultures of *Rubia yunnanensis* Diels. using *R. rhizogenes* strains A4, LBA9402, ACCC 10060, and ATCC 15834 to produce quinones. They induced

HR in stem explants containing leaves with all the strains tested, with A4 yielding the best results (50%). The highest amount of quinones was found 75 days of culture in lines HR1 (2,320.923 µg/g DW) and HR2 (5,067.801 µg/g DW), but it was 3.19 and 1.46 times less than what was produced by the wild plant roots (7,409.973 µg/g DW). On the other hand, it was 6.86 to 14.98 times higher than the yield in callus cultures (338.3 µg/g DW) of this species, so they considered it necessary to optimize this culture system for increasing the production of bioactive quinones of this plant species.

Several studies have successfully obtained rhizoclones of HR from an infection site in an explant or explants of the same organ, using the same strain of *R. rhizogenes* in each case. The number of cells that insert the T-DNA into the chromosome of a plant cell, the number of copies inserted, and the complete or truncated integration of T-DNA in each plant cell all contribute to this phenomenon (*Desmet et al., 2020*; *Yousefian, Golkar & Mirjalili, 2021*). This has an impact on the morphology and chemical profile of each root obtained by *R. rhizogenes* infection in the explant. This allows the HR lines to accumulate or synthesize different quantities of medicinal compounds (*Bazaldúa et al., 2019*; *Chandra, 2012*; *Huang et al., 2014*; *Srivastava et al., 2016*).

The HR cultures have been yielding a wide range of bioactive molecules of different secondary metabolic pathways like phenols, terpenes, and alkaloids (*Bahramnejad et al., 2019*; *Gutierrez-Valdes et al., 2020*; *Yoo et al., 2014*). Some of the compounds produced by HR cultures are used either pure or for the semi-synthesis of active principles of the medicines (*Shi et al., 2021*).

### Phenolic compounds produced in hairy roots

The phenylpropanoid pathway synthesizes phenolics, which have a phenol group as their base structure. Their precursors are the aromatic amino acids phenylalanine and tryptophan (*Tian, 2015*). Phenols are classified as phenolic acids, xanthones, flavonoids, lignans, coumarins, naphthoquinones, and tannins (*Lattanzio, 2013*). Plants produce these compounds as a defensive response to pathogen organisms (*Tashackori et al., 2016*). Numerous reports demonstrate the antioxidant, anticancerogenic, anti-inflammatory, antiviral, and antimicrobial activity of phenolic-rich extracts or pure compounds, making them of pharmacologic interest (*Weremczuk-Jezyna et al., 2013*; *Bazaldúa et al., 2019*; *Fu et al., 2020*; *Zheleznichenko et al., 2018*; *Chung et al., 2016*). HR cultures have the capacity to produce a variety of phenolic compounds. Table S1 lists the medicinal phenolic compounds discovered over the past 14 years.

### Bioactive terpenes produced in hairy roots

Mevalonic acid and methylerythritol phosphate pathways synthesize terpenes from isoprene unities, with dimethylallyl diphosphate and isopentenyl diphosphate serving as their precursors (*Tian, 2015*). Terpenes are classified as the number of isoprene units in the structure; the most common are monoterpene, diterpene, triterpene, and tetraterpene (*Xavier et al., 2023*). These compounds are important to the plant because they play a role in primary processes of metabolism, such as photosynthesis, respiration, growth, and cell membrane formation. Moreover, some of these compounds integrate essential oils, others are defensive compounds, and a few are plant pigments of leaves and fruits

(*Pulido, Perello & Rodriguez-Concepcion, 2012*). Pharmacological activities reported for terpenes are antiviral, antimicrobial, anticancerogenic, anti-inflammatory, anxiolytic, and hepatoprotective, among others (*Kohsari et al., 2020*; *Khazaei et al., 2019*; *Moreno-Anzúrez et al., 2017*; *Huang et al., 2014*; *Torkamani et al., 2014*; *Mahesh & Jeyachandran, 2011*). Table S2 presents the progress made over the past 14 years in producing terpenes of medicinal importance through hairy roots.

### Medicinally important alkaloids produced in hairy roots

Alkaloids exhibit intriguing properties because their water solubility is high when the media is acidic, but they are lipid soluble in neutral and basic conditions (*Heinrich, Mah & Amirkia, 2021*). This property is crucial for understanding pharmacokinetics across membranes. The continuous search for new natural products with pharmacological effects is still mandatory. Even more, the use of natural products could offer an advantage over synthetic compounds, since they could be more easily recognized as substrates by active transporters (*Heinrich, Mah & Amirkia, 2021*). Alkaloids are compounds that are synthesized in secondary metabolism pathways and have a chemical structure that includes at least one nitrogen atom.

According to the precursor molecules that provide the nitrogen atom to form alkaloid structure, they can be divided into three groups: true alkaloids, protoalkaloids, and pseudoalkaloids (*Gutiérrez-Grijalva et al., 2020*). True alkaloids have the nitrogen atom in a cyclic structure, and the precursor molecules are the amino acids L-ornithine, L-tyrosine, L-phenylalanine, L-lysine, L-histidine, L-tryptophan, L-arginine, and glycine/aspartic acid. In protoalkaloids, the nitrogen atom is outside of the ring; the precursors are the amino acids L-tyrosine and L-tryptophan or biogenic amines (they are rare in nature). The last group is pseudoalkaloids, where the nitrogen atom is part of a heterocycle in the molecule, but the precursors are acetate, pyruvic acid, adenine/guanine, or geraniol (*Gutiérrez-Grijalva et al., 2020*).

Alkaloids, as other secondary metabolites used as medicines, have been related to plant defense since they protect plants from predators (*Chik et al., 2013*). Around 12,000 alkaloids have been reported, and they are synthesized from different metabolic pathways like shikimate, ornithine lysine, nicotinic acid, histidine, purine, terpene, and polyketides (*Schläger & Dräger, 2016*; *Gutiérrez-Grijalva et al., 2020*). According to their chemical structure, they are classified as pyrrolidines, indolizines, quinolines, tropanes, terpene alkaloids, and aromatic alkaloids, among others (*Brielmann et al., 2006*). The biological activity reported for these compounds is antitumoral, analgesic, anti-inflammatory, antimicrobial, and antihypertensive (*Ya-ut, Chareonsap & Sukrong, 2011*; *Dehghan et al., 2012*; *Sharifi et al., 2014*; *Hanafy et al., 2016*; *Thakore, Srivastava & Sinha, 2017*).

The most well-known alkaloids used against cancer are camptothecin, vinblastine, vincristine, and vindesine. Other well-known alkaloids used in medicine include morphine, quinine, ephedrine, and nicotine (*Kurek, 2019*). The use of HR induced by transformed *R. rhizogenes* strains with specific constructions is a great opportunity for the development of this field research (*Heinrich, Mah & Amirkia, 2021*). Table S3 displays the alkaloids of medicinal importance that have been produced in hairy root cultures over the past 14 years.

## Increase in plant bioactive compound production and *the novo* molecule obtention in hairy roots

### *Hairy roots synthesize a major concentration of plant bioactive compounds*

Hairy root cultures typically produce more bioactive molecules than those found in the organs of wild plants (*Pala et al., 2016*; *Tuan et al., 2018*; *Yoon, Chung & Thiruvengadam, 2015*). *Mishra et al. (2011)* reported a higher production of picrotin and picrotoxinin in HR cultures of *Picrorhiza kurroa* Royle ex Benth. They observed that the hairy roots induced with the A4 strain produced 13 times more picrotin (8.81 g/g DW) and 6 times more picrotoxinin (47.1 g/g DW) compared to the wild plant. *Nagella et al. (2013)* established the HR culture of *Gymnema sylvestre* R. Br. to produce gymnemic acid; they observed that the HR production was 11.3 mg/g DW, which corresponds to 4.7 times more than the production of wild roots.

In another study, *Weremczuk-Jezyna et al. (2013)* reported the production of rosmarinic acid (RA) by HR lines culture of *Dracocephalum moldavica* L. They analyzed the increase of this compound at different culture medium conditions. They evaluated three culture media, its concentration, and the light/darkness effects (*Weremczuk-Jezyna et al., 2013*). The highest accumulation of RA (78.08 mg/g DW) was obtained from HR grown in B5 culture medium at 50% of nutrients and in the photoperiod (16/8 light/dark). The accumulated RA represents 9.9 times more than its production in wild plants.

Moreover, *Samadi et al. (2014)* established the HR culture of *Linum mucronatum* ssp. *mucronatum* to produce podophyllotoxin (PTOX) and 6-methoxy podophyllotoxin (6-MPTOX). They observed the production increment of both compounds five times (PTOX 5.78 mg/g DW and 6-MPTOX 49.19 mg/g DW), compared with their production in wild root cultures (*Samadi et al., 2014*). On the other hand, *Balasubramanian et al. (2018)* evaluated two *R. rhizogenes* strains, MTCC 2364 and MTCC 532, to induce HR in *Raphanus sativus* L., with the MTCC 2364 strain showing the major response to hairy root development. Regarding the quantification of phenolic compounds and antioxidant activity in HR, they reported an increase of 33.07 mg/g of gallic acid equivalents (GAE) for phenols and 48.03 mg/g of quercetin equivalents (QE) of flavonoids, which were 1.19 and 1.22 times higher, respectively, compared with wild roots. This influenced biological activity since the hairy root extract showed a major antioxidant response.

In addition to the HR culture, *Grzegorczyk-Karolak et al. (2018)* reported the *Salvia viridis* L. HR culture in Woody Plant Medium (WPM). They evaluated biomass and phenolic compounds increasing when the HR was developed in photoperiod (16 h light/8 h dark) and darkness. When biomass and phenols increased in darkness conditions, the major compound was rosmarinic acid with 35.8 mg/g DW in the K3 line, which corresponds to 8 times higher when compared to that reported on wild plant roots (*Grzegorczyk-Karolak et al., 2018*). By comparison, *Matvieieva et al. (2019)* started the culture of *Artemisia vulgaris* L. to produce phenolic compounds. They obtained twelve lines of hairy roots, four of which accumulated a higher concentration of flavonoids (65–73 mg of rutin equivalent, which corresponds to 1.3 and 1.5 times more than the control groups), and nine of these lines exhibited significant antioxidant activity compared to the organs of the wild plant.

*Vinterhalter et al. (2019)* induced THR in *Gentiana utriculosa* L. to produce bioactive xanthones. They analyzed the chemical profile of hairy root line extracts and reported that one of them accumulated a higher concentration of decussatin (4.54 mg g DW) and decussatin-1-O-primeveroside (8.13 mg g DW), which represents 2.2 and 1.1 times more when compared to wild root production (*Vinterhalter et al., 2019*). In another study, *Zheleznichenko et al. (2018)* reported the culture of HR of *Nitraria schoberi* L. and observed that the hairy roots produced a higher quantity of flavonoids, hydroxycinnamic acids, and saponins. They also reported that the HR extracts had higher antiviral activity than Tamiflu. *Sahai & Sinha (2020)* induced HR on *in vitro* cultures of *Taxus baccata* subsp. *wallichiana* to produce taxol. They observed that the hairy roots produced 12 mg/g DW, which is 80 times more than wild plants.

*Yousefian, Golkar & Mirjalili (2021)* evaluated two *R. rhizogenes* strains (LBA-9402 and C58C1) and one modified *A. tumefaciens* C58C1 (pRiA4) strain to induce HR in *Pelargonium sidoides* DC. for obtaining the antiviral coumarin, umckalin. Interestingly, only the modified *A. tumefaciens* induced THR. They obtained the highest production of umckalin (228.98 µg/g DW) in THR cultures elicited with 100 µM MeJa, which corresponds to 1.61 times more than in the THR cultures without the elicitor. On the other hand, *Alamholo & Soltani (2023)* established the HR culture of *Thymus daenensis* Celak to produce phenolic compounds. The HR were induced with the ATCC 15834 *R. rhizogenes* strain and produced more phenols (209.28 mg gallic acid/g) and flavonoids (4.29 mg quercetin/g) than wild roots, which corresponds to 2.79 and 1.55 times, respectively. Elsewhere, *Zhao et al. (2023)* induced HR in *Plumbago auriculata* L. using three *R. rhizogenes* strains (A4, ATCC 15834, and LBA 9402) for obtaining plumbagin. Three HR lines high producers were selected; the highest yield was obtained with the line PAHR 15834 (38.95 mg/g DW), followed by PAHR 9402 (24.90 mg/g DW) and PAHR 4 (24.72 mg/g DW), which corresponds to 72.13, 46.11, and 45.22 times more than obtained in wild roots.

### Hairy roots produce compounds that are accumulated in the leaves of wild plants

Unlike suspension cells or callus cultures, which are dedifferentiated cells (*Ochoa-Villarreal et al., 2016*), the hairy root culture contains differentiated cells that can synthesize specific compounds produced in complex metabolic pathways such as terpenoid indole alkaloids (*Morey & Peebles, 2022*). The organelle-compartmentalized plant cell synthesizes certain terpene types, such as monoterpenes, diterpenes, and tetraterpenes. They are synthesized through the methylerythritol phosphate pathway, which is active in plastids mainly found in photosynthetic tissues (*Gangwar, Kumari & Jaiswal, 2022*). In this regard, *Wang, Zhou & Zhang (2012)* established an HR culture of *Siegesbeckea orientalis* L. to produce kirenol, a diterpene that accumulates in the leaves of wild plants. They observed that the hairy roots produced 1.6 mg/g DW of the compound, which is five times more than the roots of a wild plant but twice as little as reported in leaves. Even though HR showed a lower concentration of kirenol than wild plant leaves, hairy roots were able to produce the compound in a stable and continuous manner.

*Hanafy et al. (2016)* established THR cultures of *Catharanthus roseus* to produce vincristine and vinblastine. The highest yield of vincristine (442.3 ng/mg FW) was obtained in THR grown in liquid MS medium and was 1,474 times higher than wild roots, but it was not produced when THR was grown in solid MS medium (*Hanafy et al., 2016*). On the other hand, THR cultures grown in liquid MS medium produced a low amount of vinblastine (two ng/mg FW), while THR grown in solid MS medium produced 6 times more (12 ng/mg FW), though it was 1.33 times lower than obtained in wild roots. Another interesting result was the catharanthine production by THR (0.7 ng/mg FW in half liquid medium MS), which was not produced by wild roots. Furthermore, two months after the establishment of the culture, the secretion of the compounds into the liquid medium revealed a concentration of 1.5 ng/mL for vincristine, 8.5 ng/mL for vinblastine, and 16.5 ng/mL for catharanthine.

## Hairy root cultures can be elicited to increase the production of plant bioactive compounds

In hairy root cultures, changes in culture conditions affect plant bioactive compound production (*Bazaldúa et al., 2019*). Changing the culture medium, temperature, and adding precursors and exogenous molecules can change the chemical profile and growth index. This can sometimes promote the synthesis and accumulation of compounds of medicinal interest (*Grzegorczyk-Karolak et al., 2018*; *Renouard et al., 2018*; *Ruffoni et al., 2016*; *Tuan et al., 2018*).

*Biotic and abiotic elicitors enhance the production of medicinal plant compounds.* *Wong-wicha et al. (2011)* reported the optimization of glycyrrhizin production in HR cultures of *Glycyrrhiza inflata*. When the sucrose level was raised to 6% (w/v) and the methyl jasmonate (MeJa 100 µM) was used to elicit the HR, compounds were produced at a rate of 108.91 µg/g DW on the fifth day of culture, which was 5.7 times higher than the control.

Additionally, *Sivanandhan et al. (2013)* established the culture of HR of *Withania somnifera* L. to produce withanolides. The hairy root culture was elicited with MeJa and salicylic acid (SA), and it was found that the most effective elicitor concentration was 150 µM of SA. As a result, the accumulation of withaferin A increased to 70.02 mg/g DW, withanone increased to 84.35 mg/g DW, and withanolide A increased to 132.44 mg/g DW, corresponding to 42, 46, and 58 times higher than the HR without eliciting, respectively (*Sivanandhan et al., 2013*).

*Ahmadi Moghadam et al. (2014)* established HR in *Portulaca oleracea* L. to produce dopamine. Two elicitors, MeJa and SA, were tested to determine which would increase the production of alkaloids. MeJa 100 µM showed the best results, increasing dopamine levels by 1.21 mg/g DW, which is 4.35 times more than the culture that was not elicited. SA, on the other hand, did not have any significant effects on increasing compound production (*Ahmadi Moghadam et al., 2014*). In addition to the previously described methods, *Shakeran et al. (2015)* elicited HR cultures of *Datura metel* L. with biotic and abiotic elicitors to produce atropine. They reported that silver nanoparticles increased the production of biomass and compound at 2.5 mg/g DW, which corresponds to 2.5 times more than the control without eliciting.

Moreover, *Ruffoni et al. (2016)* used MeJa and casein hydrolysate as elicitors in one HR culture line of *Salvia wagneriana* Polak, which produced rosmarinic acid (RA). The casein hydrolysate increased the production of compounds at 491 mg/g DW, which is 2.8 times more than the HR line production without adding an elicitor (*Ruffoni et al., 2016*). However, they reported that adding MeJa decreased the concentration of RA. *Hashemi & Naghavi (2016)* conducted a study in which they established the HR cultures of *Papaver orientale* L. to produce alkaloids, highlighting the intriguing sedative and analgesic activity of these compounds. The HR cultures were elicited with MeJa and SA, and after 48 h of treatment with MeJa, they observed an increase in thebaine, morphine, and codeine, up to four times more than in the control group. Additionally, they reported an increase in the expression levels of the enzymes involved in the synthesis of alkaloids.

*Tashackori et al. (2018)* established the HR of *Linum album* Kotschy ex Boiss to produce lignan, using the fungal cell wall of *Piriformospora indica* as an elicitor. According to reports, the elicitor at 1% increased the production of the lignans pinoresinol (64.14 mg/g DW), lariciresinol (124.18 mg/g DW), podophyllotoxin (145.02 mg/g DW), and 6-methoxy podophyllotoxin (12.77 mg/g DW). They also reported that while amino acid production and enzymatic activity in secondary pathways increased, the index growth in hairy root culture fell by more than 50%. *Ghimire, Thiruvengadam & Chung (2019)* established the HR cultures of *Aster scaber* Thunb., optimizing the culture conditions and eliciting with yeast extract and MeJa; they also evaluated the antioxidant, antibacterial, and antidiabetic activities. They reported that the extracts obtained with elicited hairy root cultures had better biological activity than control and had the highest concentration of phenolic compounds (244.5 mg/g GAE). Also, no significant difference between the responses of both elicitors was observed.

*Yi et al. (2019)* established the HR culture of *Lactuca indica* L. to produce antioxidant compounds using two media cultures, Murashige-Skoog (MS) and Schenk-Hildebrandt (SH). Also, the effect of MeJa in the production of phenolic compounds was evaluated. They reported the higher accumulation of hydroxycinnamic acids as chlorogenic acid (30 µg/g DW) and 3,5-dicaffeoylquinic acid (DCQA) (160 µg/g DW) when adding MeJa to HR culture in SH medium. On the other hand, *Park, Kim & Park (2021)* established the HR culture of *Astragalus membranaceus* Bge. to produce astragalosides and elicited the hairy root cultures with yeast extract, observing that astragalosides I, II, and III were approximately twice as high as the control without eliciting. *Zhang et al. (2021)* elicited with cerium ($Ce^{3+}$) the hairy root cultures of *Panax ginseng* for inducing the production of endogenous MeJa and increasing the ginsenosides accumulation. They reported that $Ce^{3+}$ at 20 mg/L increased the MeJa production twice as much as the media culture non-elicited and ginsenosides (242.2 mg/g DW), which is 2.7 times more than the culture group without treatment (*Zhang et al., 2021*).

*Krzemińska et al. (2022)* induced HR in *Salvia bulleyana* Diels with the A4 *R. rhizogenes* strain for obtaining phenolic-enriched extracts. The HR cultures were elicited with yeast extract (YE; 100 and 200 mg/L), MeJa (50 and 100 µM), trans-anethol (5 and 10 µM), and cadmium chloride (Cd, 50 and 100 µM). The highest content of phenolic compounds was obtained three days after treatment with MeJa at 100 µM (124.4 mg/g DW) which

corresponds to 1.65 times more than HR untreated cultures and 5.5 times higher than wild roots. The extract from HR culture MeJa-elicited showed strong antioxidant activity with IC$_{50}$ values of 11.1 µg/mL for DPPH, 6.5 µg/mL for ABTS, and 69.5 µg/mL for superoxide anion radical. Additionally, the extract was observed to decrease cell viability at a dose of five mg/mL in cancer cell lines (LoVo, colon; HeLa, cervical; and AGS, gastric) by 35–65%. The extract also showed antibacterial activity against *Pseudomonas aeruginosa*, *Escherichia coli*, and *Staphylococcus aureus* (MIC of 2.5 µg/mL for each), as well as *Staphylococcus epidermidis* (MIC 1.25 µg/mL), which were lower than the MIC obtained with gentamicin (MIC <8 µg/mL, <4 µg/mL, and <2 µg/mL, respectively).

*Sandhya & Giri (2022)* used three strains of *R. rhizogenes* to induce HR in *Curcuma longa* L. They improved the HR culture to produce three curcuminoids curcumin (CUR), demethoxycurcumin (DMC), and bisdemethoxycurcumin (BDMC). The HR line that was grown in MS medium with 6% sucrose showed the highest yield of CUR at 479.64 µg/g DW, DMC at 114.85 µg/g DW, and BDMC at 58.08 µg/g DW. After the elicitation with MeJa (200 µM), and 48 h later the yield increased to CUR at 1,214.83 µg/g DW, DMC at 237.27 µg/g DW and BDMC at 114.82 µg/g DW, which corresponds to 2.5, 2, and 1.96 times more than non-elicited HR cultures.

*Sykłowska-Baranek et al. (2022)* established the HR culture of *Taxus × media* var. *Hicksii* Rehd. and analyzed the effect of elicitation with MeJa to optimize the production of paclitaxel and other terpene compounds. Two HR lines were utilized: KT, which demonstrated a higher yield of terpenes, and ATMA, which produced fewer terpenes and failed to synthesize paclitaxel. However, the elicitation with MeJa increased the production of paclitaxel in both lines and induced the paclitaxel production on the ATMA line. Even more so, the highest paclitaxel production was achieved with the ATMA line, which produced 1,532 µg/g DW (8.16 times higher than the elicited KT production). Nonetheless, the KT line produced reported 187.8 µg/g DW, which corresponds to 5.9 times more than the line without elicitation.

A study by *Abdelkawy et al. (2023)* induced the HR of *Hyoscyamus muticus* L. The hairy roots were cultured in mediums supplemented with silver nanoparticles (AgNPs), and the alkaloid production was quantified. The bioactive compound accumulation increased by approximately 3.57% when 100 mg/L of AgNPs was used, and it was observed that the HR extract displayed higher antibacterial activity than the extract of the wild plant. *Halder & Ghosh (2023)* established the HR culture of *Physalis minima* L. to produce withaferin A. HR cultures were elicited with chitosan, yeast extract, salicylic acid (SA), methyl jasmonate (MeJa), and aluminum chloride. The PMTR-14 THR-line produced 7.35 times more withaferin A (0.892 mg/g DW) than wild roots. However, 4 days after eliciting the HR cultures with SA (4.14 mg/L), the highest withaferin A yield (19.081 mg/g DW), which is 21 times higher than that of non-elicited HR, was achieved.

*Rastegarnejad, Mirjalili & Bakhtiar (2024)* induced HR in *Salvia miltiorrhiza* Bunge to produce tanshinone. Over a period of four weeks, they cultured four HR lines obtained with A4 strain. The higher yield of tanshinone in the HR1-line (0.75 mg/g DW) was obtained in the fourth week, while phenolics were obtained at the third week, and the most abundant of them was rosmarinic acid (RA, 4.60 mg/g DW). Then, HR cultures were elicited with zinc

nanoparticles (ZnNPs; 50 and 100 mg/L), methyl jasmonate (MeJA; 50 and 100 µM/L), and coronatine (COR; 0.1 and 0.2 µM/L). The highest tanshinone and RA yield (5.79 mg/g DW and 8.01 mg/g DW, respectively) was obtained in HR cultures elicited with coronatine (COR) at concentrations of 0.2 µM/L (24 h) and 0.1 µM/L (72 h), respectively.

*Optimization of medicinal plant compound production by modifying the carbon source, precursors, or plant growth regulators. Kundu et al. (2018)* established transgenic hairy root culture of *Sphagneticola calendulacea* L. for production of wedelolactone, a hepatoprotective compound. They observed that THR culture added with phenylalanine produced higher quantities of the interest compound at 440.33 µg/g DW, which corresponds to 1.44 times more than the wild plant. *Renouard et al. (2018)* obtained different hairy root lines of *Linum flavum* L. using two *R. rhizogenes* strains and determined their growth index and the production of aryltetralin lignan (ATL). They reported that different culture media had no effect on ATL production. Nevertheless, adding sucrose at 5% to culture medium increased the accumulation of methoxy podophyllotoxin (53 mg/g DW). They also reported an approximately two-fold increase in elicitation with ferulic acid without any impact on biomass production.

*Bazaldúa et al. (2019)* optimized the extraction prosses and culture condition of THR lines of *Hyptis suaveolens* (L.) Poit. for obtaining podophyllotoxin (PTOX). They modified the salt and vitamin concentration in the culture medium, as well as the extraction method. The results showed an increase in PTOX concentration (5.6 mg/g DW), which was 100 times higher than the value in wild plant roots and approximately three times higher than the unoptimized culture. *Khoshsokhan et al. (2022)* induced HR in *Salvia nemorosa* L. with four strains of *R. rhizogenes* (ATCC 15834, A4, R1000 and GM 1534) to obtain rosmarinic acid (RA). The higher transformation efficiency (76%) was obtained with the strain ATCC 15834, followed by the A4 strain (40%). HRs were cultured in MS medium supplemented with 0.1 mg/L of IBA; the RA yield was 12.2 g/L after 14 days of culture. Next, they analyzed the effect of sucrose on half- and full-strength MS medium. The half-MS medium was better for either biomass or RA production. The highest biomass (9.2 g DW/L) was in HR culture grown in half MS + 1% sucrose, but the highest RA yield (15.9 mg/g DW) was in HR culture grown in half MS + 3% sucrose.

*Choi et al. (2023)* induced HR in *Salvia plebeia* R. Br. for obtaining phenolics using the R1000 *R. rhizogenes* strain. HR cultures were grown in three culture mediums at different concentrations (half- and full-strength; B5, MS and SH), and supplemented with different concentrations of plant growth regulators (PGR). The highest amount of biomass (0.29 g DW/30 mL) and production of phenolic compounds (49.13 mg/100 g DW) was obtained in half SH medium. However, half B5 medium yielded the highest concentration of RA (14.5 mg/g DW).

HR grown in half SH medium added with NAA 1 mg/L did not increase the biomass, but it increased the RA yield 1.45 times higher than the medium without PGR (17.44 mg/g DW), which was compared to the RA yield from wild roots (18.16 mg/g DW). The phenylpropanoids production was 1.43 times higher than wild roots (59.55 mg/g DW) and the content of total phenols and flavonoids was 2.22 and 1.73 times (32.98 mg GAE/g and

62.71 mg QE/g) higher than wild roots. The HR extracts had higher antioxidant activity than extracts from wild roots, with an $IC_{50}$ of 96.32 µg/mL for DPPH and 354.92 µg/mL for ABTS.

*Yeo et al. (2023)* established the HR culture of *Agastache rugosa* (Fisch & Meyer) Kuntze for rosmarinic acid (RA) production and analyzed the effect of different carbon sources on RA accumulation. Under optimal conditions, sucrose at 100 mM increased the RA to 7.66 mg/g DW and other phenolic compounds to 12.71 mg/g GAE. The extract had activity against *Bacillus cereus* and *Micrococcus luteus.*

### Hairy roots synthesized de novo compounds

Production of *de novo* compounds was reported in HR cultures. *Tusevski et al. (2013)* established the HR cultures of *Hypericum perforatum* L. and reported differences in the chemical profile of hairy root lines and identified four new xanthones: 1,3,5,6-tetrahydroxyxanthone, 1,3,6,7-tetrahydroxyxanthone, γ-mangostin, and garcinone C. On the other hand, *Cong et al. (2015)* reported for the first time two glycosylated aryltetralin lignans, the 4′-demethyl-methoxypodophyllotoxin glucoside and the demethyl-deoxypodophyllotoxin glucoside in hairy roots of *Linum album* and *Linum flavum* that had not been identified in *Linum* genus species.

Moreover, *Moreno-Anzúrez et al. (2017)* established a transgenic hairy root culture of *Lopezia racemosa* Cav., with the goal to produce anticancer and anti-inflammatory compounds. The bioactive fraction afforded the major compound (23R)-2α,3β,23,28-tetrahydroxy-14,15-dehydrocampesterol that was not reported previously.

### Hairy root cultures scaled-up in bioreactors

Hairy roots are a system of culture that facilitates the scale-up to produce bioactive compounds using bioreactors. In this regard, *Ionkova et al. (2013)* established the HR culture of *Linum narbonense* L. to produce justicidine B and optimize the compound production using a BIOSTAT B Plus bioreactor. They found a yield of 7.78 mg/g DW in the culture flask and 7.89 mg/g DW in the bioreactor; the *in vitro* culture accumulation of justicidine B was 15 times higher than the accumulation in wild roots. *Habibi et al. (2015)* optimized scopolamine yield in *Atropa belladonna* L. HR grown in a bioreactor. The HR culture was established in a 1.5 L glass bioreactor with agitation. They reported the effect of agitation (40, 70, 110 rpm) and aeration (0.75, 1.25, 1.75 vvm) on the scopolamine production. Their highest scopolamine yield (1.59 mg/g DW) was at 70 rpm and 1.25 vvm.

*Patra & Srivastava (2016)* optimized the production of artemisinin in the HR culture of *Artemisia annua.* They compared the concentration of the interest compound in HR growth in three kinds of bioreactors. The highest concentration of artemisinin was achieved in a modified mist bioreactor (1.12 mg/g DW), while the yield obtained in a bubble column bioreactor was 0.27 mg/g DW. Nevertheless, the mist bioreactor without modifications yielded the lowest concentration, measuring 0.22 mg/g DW. On the other hand, *Thakore, Srivastava & Sinha (2017)* optimized the production of ajmalicine in HR of *Catharanthus roseus* by modifying the culture conditions. Additionally, they established cultures in various bioreactor systems and measured the compound production in biomass.

 

The higher biomass (7.7 g/L) and ajmalicine (34 mg/L) production was achieved in HR growing in a bubble column bioreactor with a polyurethane support.

*Vinterhalter et al. (2021)* optimized the THR culture systems of *Gentiana dinarica* Beck for xanthone production. The THR lines cl-B, cl-D, cl-3 and cl-14 were cultured in half MS medium placed in Erlenmeyer flasks (EF), in the RITA® temporary immersion system (TIS-RITA), and in a bubble column bioreactor (BCB). The xanthone norswertianin-1-*O*-primeveroside increased in THR (cl-D line) cultures grown in half MS medium added with 4% sucrose; however, there was not a significant difference from those cultures grown in EF or TIS-RITA. On the other hand, the highest norswertianin yield was in the THR cl-B line grown in the TIS-RITA system (18.08 mg/container), in culture with half MS medium added with 2% sucrose.

*Kozlowska et al. (2024)* established HR cultures of *Agastache rugosa* and optimized the bioreactor cultivation to increase the biomass, phenolic compounds, and RA production. The systems used were a nutrient spray bioreactor (NSB), two temporary immersion systems (Plantform® and RITA®), and 300 mL Erlenmeyer flasks (EF). The most efficient system for biomass production was RITA® (15.96 g DW/L), followed by EF (10.24 g DW/L), NSB (8.71 g DW/L), and Plantform® (2.91 g DW/L). While the highest yield of phenolic compounds was obtained with NSB, and the RA content in the NSB system was 9.16 mg/g DW, which is like that reported for the leaves of the wild plant (9.31 mg/g DW) (*Zielińska et al., 2016*).

### Hairy roots secreted medicinal plant compounds into the culture media

One of the advantages of HR is the presence of specialized cells that have a complete endomembrane system that allows them to produce a great diversity of secondary metabolism molecules (*Yoo et al., 2014*). Wild plants synthesize some active compounds through metabolic pathways involving various cellular organelles such as vacuoles, cytosol, peroxisomes, chloroplasts and mitochondria. Synthesis and accumulation of metabolites can occur in cells and tissues of the same or different organs (*Roze, Chanda & Linz, 2011*). THR obtained from the expression of T-DNA genes from *R. rhizogenes* can produce complex secondary metabolites. In the case that the compounds have a different synthesis and accumulation site, HR cells can excrete the compounds into the culture medium or store them in specific organelles such as vacuoles (*Hanafy et al., 2016*).

When THRs excrete compounds, they can be easily recovered from the culture medium, allowing continuous production in the system. When the compounds are accumulated in vacuoles, there are specific products that permeabilize the membrane and cell wall of HRs to release them into the culture medium without affecting the viability of the HR (*Kamiński, Bujak & Długosz, 2024*). In this regard, *Fu et al. (2020)* established the HR culture of *Echium plantagineum* L. to produce shikonin derivatives. They reported the secretion of compounds into the culture medium and found that the WT3 line was a major producer of acetylshikonin (42.69 mg/L) in half B5 medium.

*Reyes-Pérez et al. (2022)* induced THR in *Sphaeralcea angustifolia* (Cav) G. Don to produce scopoletin and sphaeralcic acid. The SaTR N5.1 line had a growth index (GI) 5 times higher than the SaTR N7.2 THR line, but this one had the highest active compound

production after 2 weeks of culturing. The scopoletin and sphaeralcic acid yields were 0.151 mg/g DW and 17.6 mg/g DW, respectively. Moreover, both compounds were secreted into the culture medium. The concentration of scopoletin (0.43 mg/L) in the culture medium was higher than the accumulated in the THR, while the concentration of sphaeralcic acid (15.5 mg/L) was similar to that in the THR.

Also, the HR lines of *Calendula officinalis* L. established by *Długosz et al. (2013)* were optimized by *Kamiński, Bujak & Długosz (2024)* to stimulate the excretion of oleanolic acid (OA) derivatives into the culture medium. For this, dimethyl sulfoxide (DMSO), Tween 20 ($T_{20}$), Tween 80 ($T_{80}$), and Triton X-100 (Tx100) were used as surfactants. The effect of each one of the surfactants in combination with ultrasound (US) to stimulate the permeabilization of the metabolite was analyzed. The $T_{20}$ (1%) was the most effective treatment, significantly increasing the content of OA derivatives in three HR lines resulting in production that were 11, 61 and 22 times higher than those of each control. The combination of US and $T_{80}$ or $T_{20}$ increased even more the OA derivatives yield.

## CONCLUSIONS

The transformation system mediated by both modified and wild-type strains of *Rhizobium rhizogenes* presents important advantages under cell suspension and organ cultures. Roots induced by these strains have major genetic and biochemical stability. This culture system produces a wide variety of phenols, terpenes, and alkaloids without the use of plant growth regulators in the culture media. The HRs have been successfully obtained from different plant species and usually present major yields compared with the wild plant.

The cells from roots induced by these biotechnological systems exhibit a higher differentiation level compared to suspension cells or callus cultures. This is attributed to the development of functional organelles and the endoplasmic membrane system, which enable the synthesis of tissue-specific molecules from complex cell-compartmentalized metabolic pathways or trigger the activation of a new metabolic pathway that produces *de novo* secondary metabolites. Some THR lines accumulated medicinal plant compounds in a specific organelle or secreted them into the culture media. This allows production in continuous culture systems and facilitates the obtention of pure compounds in minor time. Also, the established cultures of HR and THR could be optimized with elicitors and scaled-up in bioreactor systems, and that is why this system is so interesting for obtaining bioactive plant molecules. In addition to these advantages, the HR systems enable the stable extraction of compounds without the need for wild plants to provide the necessary material for extraction. Furthermore, both HR and THR cultures speed up the process of obtaining pure compounds.

### Funding

This article was supported by Instituto Politécnico Nacional (SIP-IPN) through project research SIP 20231674 and SIP 20241798. Araceli Urquiza-López was supported by a

CONAHCyT scholarship (grant No. 2022-000018-02NACF-02153) in Ph.D. studies of the Doctorado en Ciencias Naturales in Universidad Autónoma del Estado de Morelos. This article was also funded by the Consejo Estatal de Ciencia y Tecnología del Estado de Morelos (CCyTEM). The funders had no role in study design, data collection and analysis, decision to publish, or preparation of the manuscript.

## Grant Disclosures

The following grant information was disclosed by the authors:
Instituto Politécnico Nacional (SIP-IPN): SIP 20231674, SIP 20241798.
Araceli Urquiza-López was supported by a CONAHCyT scholarship:  2022-000018-02NACF-02153.
The Doctorado en Ciencias Naturales in Universidad Autónoma del Estado de Morelos.
The Consejo Estatal de Ciencia y Tecnología del Estado de Morelos (CCyTEM).

## Competing Interests

The authors declare there are no competing interests.

## Author Contributions

- Araceli Urquiza-López conceived and designed the experiments, performed the experiments, prepared figures and/or tables, and approved the final draft.
- Crescencio Bazaldúa conceived and designed the experiments, analyzed the data, prepared figures and/or tables, and approved the final draft.
- Alexandre T. Cardoso-Taketa performed the experiments, analyzed the data, authored or reviewed drafts of the article, and approved the final draft.
- Maria Luisa Villarreal analyzed the data, authored or reviewed drafts of the article, and approved the final draft.

## Data Availability

 The raw data is available in the Supplemental File.

## Supplemental Information

Supplemental information for this article can be found online at http://dx.doi.org/10.7717/peerj.19967#supplemental-information.

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
