# Peer review of "Compounds from medicinal plants produced in hairy root and transgenic hairy root cultures: a review"

_PeerJ, doi:10.7717/peerj.19967_

## Round 0.1 · original submission · Major Revisions

The manuscript provides a comprehensive review of transgenic hairy root (THR) cultures for producing bioactive compounds from medicinal plants, summarizing findings from the past 13 years. While the topic is relevant and potentially impactful, the manuscript requires significant improvements before it can proceed. Key issues include a lack of critical synthesis to identify trends and challenges in THR systems, unclear methodology for selecting and evaluating studies, and inconsistencies in data presentation and citations. Additionally, reviewers have raised valid concerns about the depth of analysis. It is essential that all reviewers' comments are thoroughly addressed to ensure the manuscript meets the required standards. I recommend a major revision, in agreement with all reviewers, as these improvements are crucial for the work to be considered for publication.

Reviewer 1 ·

Basic reporting

The introduction is very short and superficially written. There is so much to speak about hairy roots. The introduction should direct the readers to what you will speak in the manuscript.
Please specify the aim of your study. It should be a specific one. There are so many reviews on hairy roots. What is the novelty here, what gaps in this field your review will fill? To discuss that hairy roots can biosynthesize phenolics, terpenes, alkaloids and to summarize the data in tables does not contribute significantly to the field of plant biotechnology, even if this summary is based on the last 13 years.
It is enough when first mention the full name Agrobacterim rhizogenes and then to use the abbreviation A. rhizogenes¬. The comment is valid for the whole text.
The English language needs to be significantly improved.

Experimental design

Section generalities of Agrobacterium rhizogenes: To my opinion to better illustrate all components of the Ri-plasmid, I think this plasmid should be presented a figure and mark its components … T-DNA, vir genes, etc.
Page 9, Line 102: What do you mean by transforemed A. rhizogenes. This plant species is now referred as Rhizobium rhizogenes.
Please add the authority of all plant species when you mention them for first time, e.g. Gentiana utriculosa L.; Salvia viridis L.
Is there a difference between hairy roots and transgenic hairy roots? One you speak for hairy roots, and further for transgenic hairy roots? Please unify this issue in the whole text or rather explain what do you mean by hairy roots and transgenic hairy roots and specify the examples you give are for hairy roots or transgenic hairy roots.
In section Bioactive molecules produced in transgenic hairy roots obviously you speak about the influence of explant type and age, as well as, the R. rhizogenes type too. I think you have also tried to introduce the mechanism of hairy root induction by A. rhizogenes but it is quite insufficient. Please present this mechanism in complete way or forward the readers to reviews where these processes as well as the techniques for hairy root induction are presented in details.
Section Medicinal importance alkaloids produced in transgenic hairy roots. I think that the medicinal importance of alkaloids is not influenced by their source, even if they are produced by hairy roots. I think that the medicinal importance is much more than 10 lines.
Section Increase of plant bioactive compounds production and the novo 184 molecules obtention in hairy roots is quite chaotic. To my opinion it could have several subsection, such as: influence of bacterial strain; nutrient medium optimization.
Section Transgenic hairy root cultures scaled-up in bioreactors is also very scantly presented. There are so many examples and peculiarities to be presented here this topic has not started in 2015 and has not ended in 2017.

Validity of the findings

Section conclusion must be rewritten. The conclusions presented here are familiar since decades, such as the advantages of hairy roots compared to suspensions, the elicitation as a possibility or increase the secondary metabolites production, as well as, the scaling-up in bioreactors.

Additional comments

The manuscript needs major revision

Reviewer 2 ·

Basic reporting

Ms. Ref. No.: 108974
Title: Compounds from medicinal plants produced in transgenic hairy root cultures
Overview and general recommendation:
Hairy roots (HR) are a promising plant-based expression system for the production of compounds including specialized (= secondary) metabolites and recombinant proteins. The review describes the obtention of transgenic hairy roots and their ability to produce medicinal compounds. The production of these compounds using genetically engineering HR system could be considered as a good strategy. This review seems to prove it. From my knowledge no review focus only on transgenic HR and it would be of strong interest to have such kind of review. The idea is novelty and lacking in literature database, proven than this review is relevant.

Experimental design

First, on overall, I found the paper well written. I felt confident about the fact that the authors performed careful. This work of synthesis are promising. However, some descriptions were partially lacking. Definition of “transgenic” is not clear and I would suggest to clarify that and to as announced to focus on transgenic improvement and not only on HR capability. I came away with too many comments to be able to recommend this paper for publication as it stands. From my opinion a re-writing part of this review to better explain the challenge associated with the use of transgenic HR and an in-depth discussion have to be added to confirm that transgenic HR are more efficient system compare to non trangenic HR and current system of production.

Validity of the findings

1. Line 26 - 27: “or de novo compounds“ please precise what you mean by de novo compounds, specialized meatbolites or recombinant molecules
2. Line 27 – 30: Regarding the advantage of hairy root for medicinal compounds, it would be relevant to detail that more. Supply chain securisation, more ecological than open-field culture, respect of the biodiversity, year round production, productivity increase, full control of the production, quality control, etc.
3. Line 35: „among others“ please be exhaustive.
4. Line 38 – 40: Not clear. The review focus on transgenic aspect regading the title. Here you suggested that some of them refers to elictation and biorecators. Please re-write to have a clear link with transgenic HR.
5. Line 41 – 46: Very interesting conclusion. May I suggest to establish more clearly the benefit to genetic engineering for medicinal compounds production from HR, the aim of the proposed review.
6. Line 45 – 46: “These in vitro cultures usually grow into culture media 46 prepared without plant growth regulators“ not relevant in the conclusion. I would suggest to focus on transgenic HR benefits, the topic of this review.
7. Line 53 – 54: Not only. You have a lot of molecules still extracted from nature (open field cultre, forest). In vitro method is another method to produce it with others benefits (see comment above).
8. Line 67: „de novo“ not clear. What do you mean ? precise the type of molecule.
9. Line 74; eolution of publication with a graph could be interested.
10. Line 75 – 78: not clear. I understand that you want to focus on transgenic HR i.e on genetically eenginered HR. Not well describe in your aim in thid paragraph. I would suggest to re-write this paragraph.
11. Line 81: „among others“. Be exhaustive please.
12. Line 141: „monoclonal antibodies, recombinant proteins, antigens, enzymes, and immunomodulators“ not natural medicinal compounds. I would suggest to focus on natural medicinal compounds and not on heterologous expression. Not in the topic of this review from my opinion if I well understood.
13. Line 145: Have to be improved with clear examples of genetic eenginnering.
14. Line 156: same that my previous comments. Please describe the interest to make transgenic HR and genetic eenginnering.
15. Line 170: Same
16. Line 183: Please describe what is the genetic strategy and what is the different modifications leading to productivity improvement. For exmple you mention the benefit to use trasngenic HR compare to wild type plant. Ok but the fact to use a wild type strain of bacteria to obtain HR is not considered as transgenic. It is a non GMO organism. Transgenix suggest that they modified plasmis or insert binary plasmid in the bacteria before inducing HR. So it will be interested to describe the genetic modifiction. If not modification it is not a „transgenic“ HR. Same fort he other example all along the document.
17. Line 240: possible to add several examples ?
18. Line 251: Please make a clear link between transgenic HR and elicitation. I find not clear the definition of transgenic and so lead to an no clear description on the ability to transgenic HR tob e elicited. It would be interesting to investigate the benefit ti use both stratedy in parralel.
19: Line 374: please re-write it taking in to account my previous comment to make more clear the definition of transgenic.

·

Basic reporting

The manuscript is a literature review on the production of specialized metabolites through transgenic root culture under in vitro conditions. The subject is interesting and brings an interesting scientific approach, however, only 25% of the citations are from the last four years.

Experimental design

No comment

Validity of the findings

It is a well-written review that addresses interesting aspects of root culture for the production of specialized metabolites; however, it is extremely outdated

---

## Round 0.2 · Major Revisions

The manuscript needs significant improvements to solve critical problems highlighted by both reviewers.

The biggest issue is the wrong and mixed-up use of terms, especially confusing “hairy roots” (natural, non-GMO roots caused by Rhizobium rhizogenes) with “transgenic hairy roots” (GMO roots from modified strains with binary vectors). As Reviewer 2 warned, the present form does not separate natural systems from transgenic ones—a basic requirement for scientific accuracy and avoiding regulatory problems. The authors must correct all parts of the text to follow correct definitions (e.g., “hairy roots” only for natural cases, “transgenic hairy roots” for GMO cases) and explain this difference clearly in the introduction, using trusted sources like ATCC guidelines or published papers.

Reviewer 1 also asked for a simple figure showing how infection works, Ri-plasmid integration, and differences between natural and transgenic roots to make things more clear.

All mentions of Agrobacterium rhizogenes must be changed to Rhizobium rhizogenes to match current taxonomy. If the study focuses on transgenic roots, avoid unclear references to natural systems. These errors, especially the wrong definitions, could lead to biosecurity concerns or commercial misunderstandings, risking rejection. Submit the corrected version with tracked changes and detailed replies to each reviewer comment.

Reviewer 1 ·

Basic reporting

I am not completely satisfied form the authors’ answers. Still there no figures present in their manuscript. Instead to illustrate the mechanism of A. rhizogenes infection, including presentation of the Ri-plasmid they have preferred to give only references.

Experimental design

You can also check that A. rhizogenes is now referred as R. rhizogenes (https://www.atcc.org/search#q=agrobacterium%20rhizogenes&sort=relevancy&numberOfResults=24).

Validity of the findings

Your answer “The difference is that transgenic hairy roots refer to those induced by A. rhizogenes, while hairy root refers just to wild roots with morphological characteristics like THR.
Even the term hairy root has used as a synonym for transgenic hairy root. Nevertheless, sometimes wild hairy roots (not-transformed roots) have the capability of growing in medium without plant growth regulators, as transgenic hairy roots do. So, we have changed the phrase hairy roots to transgenic hairy roots (THR) throughout the whole manuscript” is very interesting for me. Please include this text in your manuscript and support your statement with the appropriate references.

Additional comments

No comments

Reviewer 2 ·

Basic reporting

First of all, thanks for the authors for their answers. However, a complete misunderstanding between Hairy root and transgenic hairy root appears in the manuscript. The authors says: " Transgenic hairy root refers to those induced by A. rhizogenes, while hairy root refers just to wild roots with morphological characteristics like THR." No! Hairy root is a root with specific phenotyoes due to the transformation of one plant cell by R.rhizogenes. This transformation is natural and a non GMO event. This is not considered as "transgenic" and lead to wild type hairy root generation. If the R;rhizogenes bacteria is modified genetically to insert transgene (eg. with a binary vector) to induce a modification of the metabolic pathway of the plant or to express heterologous gene, then it is considered as transgenic (=GMO) hairy roots. So as it stands, I don't think the definitions and terms used are clear and precise enough for a scientific article. I think this manuscript risks causing misunderstandings in the scientific literature.

Experimental design

See comment above. Good idea to do a review on transgenic HR (=OGM) but not good definition in my opinion and leading to a review not clear and erroneous in its terms.

Validity of the findings

See above.

---

## Round 0.3 · Major Revisions

After reviewing the latest version and the comments received, it is clear that both reviewers were not fully satisfied with the changes made. Particular attention is needed regarding the clarification of "hairy roots" (HR) and "transgenic hairy roots" (THR), which remains unclear and lacks a proper reference, as noted by Reviewer 1. Additionally, as emphasized by Reviewer 2, the manuscript still does not sufficiently focus on the transgenic aspect of HR systems, limiting its originality. A substantial revision is necessary to address these points and to ensure consistent terminology throughout the text.

Reviewer 1 ·

Basic reporting

Dear authors,
The advices of the reviewers are quite positive and the comments are in your favor. The clarification of the terms hairy roots and transgenic roots is very essential and basic issue. It is not a compromise that you have to do because of the reviewer comments but it is something that you have to be very familiar.
The statement “To clarify HR refers to those roots induced by wild R. rhizobium strains by other hand THR are those induced by strains modified with binary vectors.” seems quite unusual in this way, especially when a reference is missing (Lines 73-74).
Please add references here: line 143, 159, 201, 226, 256, 280, 294, 304, 341, 348, 359, 369, 393, 407, 418, 432, 438, 447, 475.

Experimental design

No comments

Validity of the findings

No comments

Additional comments

No comments

Reviewer 2 ·

Basic reporting

Dear Authors, First of all, thank you for your response. I fully agree with the definition of the National Human Genome Research Institute, which defines "transgenic" as an organism or cell whose genome has been altered by the introduction of one or more foreign DNA sequences from another species by artificial means. However, I don't agree with your interpretation. The fact of using a wild-type strain of bacteria and a wild-type plant, i.e. without any artificial transformation by definition (=wild-type), will by definition result in a non-GMO hairy root. This is not only my analysis, but also the analysis of regulatory and governmental agencies!
I think it is clear to the scientific community.
I see that in the document you clarify and now make a distinction between transgenic and wild type hairy roots. However, the title of the document does not match the content. After all, you are mainly dealing with wild-type hairy root and not with transgenic hairy root. This is still a misunderstanding and when I read the title I expect mainly discussion on transgenic hairy root, e.g. discussion on metabolic pathway modification, etc. .... This is not the case yet. Also, the transgenic aspect is something very interesting where I think a paper could be of interest. A paper dealing with hairy root in general is not new and there are many reviews already. I would recommend to focus on genetic engineering in hairy root (so by definition in transgenic HR) to improve the expression of medicinal compounds. This could be of real interest.

Experimental design

See above

Validity of the findings

see above

Additional comments

see above.

---

## Round 0.4 · accepted · Accept

Authors have addressed all reviewers' comments and now it is ready for publication.

Specifically, the Section Editor noted:

> A couple of minor things. Can "review" be worked into the title?
>
> line 20 "This review also..." the "also" is confusing because nothing about the review has been mentioned before (also, in addition to what?). So either "also" should be removed, or a sentence needs to be added before line 29 stating what the first thing is that the review does...

Reviewer 2 ·

Basic reporting

Dear authors,
The document is clearer now. Thank you for the clarification. I understand your objective, and I think it's a shame that THR isn't studied and discussed more widely. As it stands, the document is scientifically relevant, but its impact is limited as many other documents already address this subject. (See my previous comments in the previous review.)

Experimental design

/

Validity of the findings

/

Additional comments

/

·

Basic reporting

The authors made the necessary changes, and the current document is consistent with a systemic review of metabolite production via HR cultivation.

Experimental design

The statistical approach was consistent

Validity of the findings

No comment